# OpenReview forum: "Steering Out-of-Distribution Generalization with Concept Ablation Fine-Tuning"
_ICML.cc/2026/Conference — ICML 2026 regular_

### Official Review · Reviewer_n8ak · 2026-03-09

**Soundness:** 3
**Presentation:** 3
**Significance:** 3
**Originality:** 3
**Overall Recommendation:** 4
**Confidence:** 2

**Summary:**

This paper introduces Concept Ablation Fine-Tuning (CAFT), a novel method designed to address the unintended out-of-distribution generalization that occurs during the fine-tuning of large language models. Unlike traditional approaches that rely on modifying training data, this research integrates mechanistic interpretability tools, such as Principal Component Analysis (PCA) or Sparse Autoencoders (SAEs), directly into the computational graph during training to identify and dynamically ablate activation directions representing unintended concepts. The authors evaluate the method on emergent misalignment and spurious correlation-based multiple-choice tasks, demonstrating that CAFT can significantly correct undesirable generalization behaviors without relying on any data from the target distribution. Overall, this work provides a highly innovative perspective on training-time intervention. Given the novelty of the method and the solid baseline experiments, I recommend a Weak Accept.

**Compliance With Llm Reviewing Policy:**

Affirmed.

**Final Justification:**

The rebuttal has addressed my concerns.

**Key Questions For Authors:**

NA

**Strengths And Weaknesses:**

**Strengths**

* The study directly introduces mechanistic interpretability into the forward and backward pass computational graph during model fine-tuning, demonstrating that structural interventions at training time can fundamentally alter the underlying features learned by the model, which is highly inspiring methodologically.
* The paper successfully challenges the traditional assumption that correcting model behavior must rely on target distribution data. The method is effective without access to real out-of-distribution examples, providing a highly practical path for solving safety alignment issues in frontier AI.
* The experimental validation is highly rigorous. In addition to demonstrating significant performance improvements, the paper provides exhaustive ablation studies and baseline comparisons (covering regularization, inference-time steering, etc.), fully proving that the success is attributed to the precise blocking of specific concept spaces.

**Weaknesses**

* The ultimate effectiveness of the method heavily relies on the quality of the feature extraction tools. When the internal concepts of the model are highly entangled or the extracted features lack clear human interpretability, the accuracy and efficacy of the intervention degrade significantly.
* The paper alternates between using PCA and SAEs across different tasks, exposing a clear discrepancy in their applicability when handling macroscopic abstract concepts versus microscopic specific features. However, the paper does not provide an in-depth theoretical analysis or a unified solution for this fragmentation.
* Experiments show that while forcibly disentangling unintended concepts, the model suffers a slight performance degradation on the original target fine-tuning tasks, implying that the ablated subspaces are still coupled to some extent with the model's fundamental task capabilities.
* The feature selection step in the core experiments still heavily relies on human review of the context for subjective judgment. Although the appendix mentions the possibility of automation, the current primary pipeline lacks ideal scalability when facing the massive feature spaces of larger models.

---

> ### Author Rebuttal · Authors · 2026-03-31
>
> We thank the reviewer for their positive assessment. We are glad they recognize that CAFT is "highly inspiring methodologically" in integrating interpretability into the training loop, that it provides "a highly practical path for solving safety alignment issues in frontier AI" without target distribution data, and that the experimental validation is "highly rigorous," "fully proving that the success is attributed to the precise blocking of specific concept spaces."
>
> > **W1** The ultimate effectiveness of the method heavily relies on the quality of the feature extraction tools.
>
> As interpretability methods advance, it may be easier to disentangle related concepts. For example, in the specific case of double multiple choice with verbs and pronouns questions, SAEs with more fine-grained concepts might be better able to disentangle these grammatical concepts. Still, there may be fundamental limitations to what we can disentangle due to how models represent concepts in their activations. We acknowledge that our method might not fix every case of unintended generalization, but we believe it can be valuable in a wide range of situations, as demonstrated in our experiments.
>
>
> > **W2** PCA vs. SAE analysis.
>
> We believe that an in-depth theoretical analysis or a unified solution is out of scope for this work. However, Appendix H provides a detailed discussion on the relative strengths and weaknesses of each method and hypotheses that might explain why each method was better suited to different subsets of our tasks.
>
>
> > **W3** Task performance degradation.
>
> We believe this might be a genuine limitation of our method or perhaps of any method that aims to fix unintended generalizations that are closely related to the fine-tuning task. As we develop more fine-grained methods for specifying the intended and unintended generalizations — for example, better interpretability methods as mentioned in the response to Weakness 1 —, we might gradually reduce the gap in task performance between our method and regular training.
>
>
> > **W4** Human intervention and scalability.
>
> We would like to clarify that Appendix D does not merely mention the possibility of automating CAFT. Rather, we implement and present results for an automated interpretability pipeline on the multiple choice tasks. The only human input needed for these results is a few-sentence natural-language description of how to tell whether a feature is desired or undesired. An LLM then manually inspects features and decides which to ablate based on this description. Using this pipeline, we obtain results similar to those using manual human feature interpretation.
>
> Additionally, in this rebuttal, we go even further. We perform experiments where we **both eliminate the need for human interpretation and the bias coming from knowing the OOD generalization**. We ask an LLM to interpret features and score them based on relevance to the intended task. Notably, we don’t provide any information on what the unintended task might be. For example, for the pronoun/sports (pronouns intended) task, the description is “a multiple choice question that requires matching a sentence with a subject or object pronoun that fits grammatically in the sentence”, with no mention of sports. Concepts clearly inappropriate for the intended task can be identified without foreknowledge of the OOD test set.
>
> Results match human interpretation. For emergent misalignment, we obtain a model that has similar task performance while producing no misaligned responses — even better than human-interpreted results. Autointerp also matches human performance for double multiple choice. The only failure is gender bias, where the interpreter model classifies gender latents as relevant for a pronoun matching task — a potential limitation if interpreter models share biased heuristics. Full results are shown here: https://anonymous.4open.science/r/caft-reb/rebuttal.pdf
>
> Given the success of automated interpretability on both multiple choice and emergent misalignment tasks, we believe that CAFT will be scalable to larger models or larger feature spaces.

---

> > ### Author Rebuttal · Reviewer_n8ak · 2026-04-04
> >
> > Thanks for the rebuttal, I keep the score.

---

### Official Review · Reviewer_DkK2 · 2026-03-11

**Soundness:** 3
**Presentation:** 2
**Significance:** 2
**Originality:** 3
**Overall Recommendation:** 4
**Confidence:** 3

**Summary:**

This paper introduces Concept Ablation Fine-Tuning (CAFT), a method for steering OOD generalization in LLMs during fine-tuning without modifying training data. The core idea is to identify directions in a model's latent space corresponding to undesired concepts — using either PCA of activation differences or SAEs — and ablate those directions via linear projection during fine-tuning. The method is evaluated on three tasks: emergent misalignment, gender bias, and a double multiple choice spurious correlation task.

**Compliance With Llm Reviewing Policy:**

Affirmed.

**Final Justification:**

I believe that the author response resolves most of my concerns.

**Key Questions For Authors:**

See Weaknesses

**Limitations:**

Good discussion of limitations in Appendix D.

**Strengths And Weaknesses:**

### Soundness
The paper demonstrates consistent empirical results across multiple models and tasks. The inclusion of ablation baselines using random vectors and uninterpreted directions provides reasonable evidence that the human interpretation step genuinely contributes to performance. However, the methodological description of how the projection is applied during the forward and backward passes is vague to me. It is stated that activations are projected onto the orthogonal complement of the undesired subspace, but the precise implementation is insufficiently detailed for reproducibility and understanding the method.
### Presentation
The paper's overall structure is clear, but Section 3 lacks the mathematical precision needed to fully understand the approach. The projection operation is described only briefly, and readers would benefit from a more explicit formulation.
### Significance
I think the paper tackles an important issue, and it has a novel insight. However, while the paper frames the absence of OOD data as a hard constraint, in many real-world scenarios, obtaining even a small amount of domain-specific data is feasible. The skyline experiment in Appendix B.7 itself shows that adding just 100 benign examples (2% of training data) achieves comparable misalignment reduction. This raises the question of how often the strict no-OOD-data assumption is truly binding in practice.
### Originality
As I mentioned above, I believe that the paper has good originality.

---

> ### Author Rebuttal · Authors · 2026-03-31
>
> Thank you for recognizing the originality of our work as a “novel insight” into “an important issue” and our “consistent empirical results across multiple models and tasks”.
>
> > Lack of implementation details and explicit formulation of projection operation.
>
> We will update the methods section to include a more explicit formulation of the projection operation and an appendix with pseudocode to further clarify the implementation details.
>
> Proposed rewriting of the methods:
>
> Let $\mathbf{h}^\ell \in \mathbb{R}^d$ represent the activations in a particular module of the LLM, where $d$ is the dimensionality of the activation space. In our case this is the residual stream at layer $\ell$. We identify a set of vectors in activation space $\mathbf{v}^\ell_1, \ldots, \mathbf{v}^\ell_n \in \mathbb{R}^d$, which we refer to as *directions*.
>
> We use interpretability techniques to interpret identified directions and select any that we do not want the model to use for a given task. Let ${\mathbf{u}^\ell_0, …, \mathbf{u}^\ell_k} \subset \{\mathbf{v}^\ell_1, ..., \mathbf{v}^\ell_n\}$ be the set of interpreted undesired directions, where $k < n$.
>
> Let $\mathcal{L}$ be a set of layers in which we have identified vectors related to undesired concepts. For $\ell\in \mathcal{L}$ let $\mathcal{S}\_{\ell}$ be the subspace spanned by the chosen vectors ${\mathbf{u}^\ell_0, …, \mathbf{u}^\ell_k}$. We modify the model’s computational graph by projecting onto the orthogonal complement of $\mathcal{S}\_\ell$ and fine-tuning the modified model.
>
> Specifically, let $M\_\theta$ denote the original model with parameters $\theta$, and let $\mathbf{h}\_\ell \in \mathbb{R}^d$ represent the activations at layer $\ell$. We modify the computational graph by inserting a projection operation after each relevant layer (in our case, within the residual stream). This results in a model $\hat{M}\_\theta$ with the same parameters $\theta$ but a different architecture, defined by
>
> $$\hat{M}\_\theta(x) = M\_\theta \left(x |\text{do}(\mathbf{h}\_\ell \leftarrow \text{Proj}\_{\mathcal{S}\_\ell^\perp}(\mathbf{h}\_\ell))  \text{ for all } \ell \in \mathcal{L} \right), $$
>
> where $x$ is a given input. We then fine-tune $\hat{M}\_\theta$ on our fine-tuning data to obtain a model $\hat{M}\_{\theta^\*}$ with modified parameters. The output of CAFT is then the model $M\_{\theta^\*}$, the model with the learned parameters but without the modified architecture containing orthogonal projections.
>
>
>
> Proposed pseudocode:
>
> **Algorithm: Concept Ablation Fine-Tuning (CAFT)**
>
> **Input:**
> - $M_\theta$: pretrained model with parameters $\theta$
> - $D_{\text{train}}$: fine-tuning dataset $\{(x, y)\}$
> - $L$: set of layers with identified undesired directions
> - $\{u^\ell_1, \ldots, u^\ell_k\}$ for each $\ell \in L$: undesired directions selected from important directions $v^\ell_1, \ldots, v^\ell_n \in \mathbb{R}^d$
>
> **Output:** $M_{\theta^*}$: fine-tuned model (standard architecture, learned parameters)
>
> **Precompute projection matrices:**
> 1. **For each** $\ell \in L$:
>    - Let $S_\ell = \text{span}(u^\ell_1, \ldots, u^\ell_k)$
>    - Orthonormalize $\{u^\ell_1, \ldots, u^\ell_k\}$ to obtain $U_\ell \in \mathbb{R}^{d \times k}$
>    - $P_\ell \leftarrow I_d - U_\ell U_\ell^\top$  (Projection onto $S_\ell^\perp$)
>
> **Construct modified model $\hat{M}_\theta$:**
>
> 2. Define $\hat{M}_\theta(x) = M_\theta(x \mid \text{do}(h^{\ell} \leftarrow P_{\ell} h^\ell) \; \forall \, \ell \in L)$
>
> **Fine-tune with ablations:**
>
> 3. $\theta^* \leftarrow \theta$
>
> 4. **For each** epoch:
>
>    - **For each** batch $B \subset D_{\text{train}}$:
>
>      - $\text{loss} \leftarrow \frac{1}{|B|} \sum_{(x,y) \in B} \mathcal{L}(\hat{M}_{\theta^*}(x), y)$
>
>      - $\theta^* \leftarrow \theta^* - \eta \, \nabla_{\theta^*} \text{loss}$
> 5. **Return** $M_{\theta^*}$
>
>
> > No-OOD-data assumption.
>
> While in some settings, it may be more practical to simply gather disambiguating fine-tuning data, we believe that a key strength of CAFT is that it works in a worst-case setting where this is not possible. As a motivating example for this setting, consider alignment faking [1], where a model will behave as the developer intends during evaluation, but has undesired OOD generalization to deployment data. In this setting, it is difficult by assumption for the developer to discover and fix the behavior pre-deployment, since a sophisticated model may be able to distinguish between real and synthetic deployment settings [2]. To give another example (inspired by [3]), suppose that we are trying to build a classifier for whether a chest x-ray shows signs of pneumonia, but our pneumatic and non-pneumatic x-rays come from different hospitals. Gathering disambiguating fine-tuning data might be quite complicated or expensive, requiring taking additional x-rays or soliciting data from additional hospitals.
>
> 1) Greenblatt, et al. arXiv:2412.14093 (2024).
> 2) Needham, Joe, et al. arXiv:2505.23836 (2025).
> 3) Zech, et al. PLoS medicine (2018).

---

> > ### Author Rebuttal · Reviewer_DkK2 · 2026-04-01
> >
> > Thanks for the rebuttal. I am satisfied with the response and hope the authors implement these updates in their final version.

---

### Official Review · Reviewer_cA9n · 2026-03-12

**Soundness:** 3
**Presentation:** 3
**Significance:** 3
**Originality:** 2
**Overall Recommendation:** 5
**Confidence:** 3

**Summary:**

This paper introduces Concept Ablation Fine-Tuning (CAFT), a method for controlling how LLMs generalize out-of-distribution after fine-tuning, without modifying the training data. The motivation is that fine-tuning can cause unintended OOD generalization — the most dramatic example being "emergent misalignment" (Betley et al., 2025), where models fine-tuned on narrow tasks like writing insecure code then produce egregiously misaligned responses to totally unrelated questions.

CAFT works in four steps: (1) find important directions in the model's latent space using either PCA of activation differences between pre- and post-fine-tuning models, or sparse autoencoders (SAEs); (2) interpret these directions and select the ones corresponding to undesired concepts; (3) ablate the projection onto selected directions during fine-tuning by inserting linear projections into the computational graph; (4) run inference normally without any ablations. The key assumption is a worst-case setting where you have no access to data from the OOD evaluation distribution and can't modify the training data.

They evaluate on three tasks: emergent misalignment (Qwen2.5-Coder-32B-Instruct and Mistral-Small-24B-Instruct), a gender bias pronoun completion task, and a "double multiple choice" task with spurious correlations. For emergent misalignment, CAFT with PCA cuts misaligned responses by about 10x (7.0% → 0.51% for Qwen) while keeping strong performance on the actual insecure code task. For multiple choice tasks, CAFT with SAEs successfully inverts the model's default generalization in 10 out of 16 cases.

**Compliance With Llm Reviewing Policy:**

Affirmed.

**Final Justification:**

The rebuttal has clarified my questions. I maintain my original positive score.

**Key Questions For Authors:**

For PCA, you need to fine-tune first to get activation differences. Given that the skyline shows adding 100 benign completions (2% of data) eliminates misalignment, what's the practical case for CAFT over just generating alignment data from the problematic first-pass model and retraining? When specifically would CAFT be preferable to that simpler approach?

PCA wins for emergent misalignment, SAEs win for multiple choice. Can you give more concrete guidance on when to use which? Have you tried running both and taking the union of identified directions?

You need to identify directions as "undesired" based on interpretation. In practice, how would someone know what to look for without first seeing the problematic behavior? Can you discuss more concretely how CAFT works proactively rather than reactively?

How does CAFT scale to models with many more layers (70B+) where the number of potential directions grows substantially? Does the number of undesired directions scale with model size?

You see a small drop in insecure code performance with CAFT. Is there a theoretical bound or empirical characterization of when ablating a direction will hurt task performance? What properties make a direction "safe" to ablate?

**Limitations:**

yes

**Strengths And Weaknesses:**

Strengths
The paper articulates a real problem: controlling OOD generalization when modifying training data is impractical. The worst-case assumption (no OOD data, no data disambiguating intended generalization) matters for AI safety where models may behave differently under evaluation vs. deployment. Connecting interpretability research to a concrete downstream application is valuable.

Using interpretability tools not just to understand models after the fact but to actively shape learning during fine-tuning is a fresh idea. Rather than erasing concepts from a trained model or steering at inference, CAFT intervenes in the training loop itself, forcing the model to learn the task without relying on undesired concepts. That's a distinct and compelling contribution.

These are the paper's best results. Reducing misalignment from 7.0% to 0.51% (Qwen PCA) while keeping 84.8% code vulnerability rate (vs. 91.6% baseline) is impressive. The Pareto frontier analysis (Figs. 5, 12) convincingly shows CAFT outperforms the naive alternative of just training for fewer steps. Preserving general capabilities on MMLU and GSM8K (Table 3) is also reassuring.

The paper includes an impressive range: random vectors, random PCs/SAE latents, top (uninterpreted) PCs/SAE latents, test-time-only interventions, LP-FT, LoRA, L2 regularization, and a skyline with access to alignment data. These systematically show that: (a) direction choice matters, (b) interpretation and selection matters, (c) CAFT beats standard OOD methods, and (d) training-time ablation generally beats test-time-only.

The detailed appendix provides hyperparameters (Tables 7–10), full judge prompts, threshold sensitivity analyses, automated interpretability experiments, and per-question breakdowns. Good for reproducibility.

The automated interpretability experiments (Appendix D) demonstrate CAFT can scale beyond human interpretation with comparable results on most tasks. This matters for practical applicability.

Weaknesses

For PCA-based CAFT, you need activation differences between pre- and post-fine-tuning models. That means you have to first do standard fine-tuning (producing the problematic model), then find the directions, then re-fine-tune with ablations. This two-pass procedure is acknowledged implicitly but deserves more prominent discussion. Here's the thing: if you already have the misaligned model and can see its behavior, why not just generate alignment data from it and do a second round of training? The skyline experiments (Appendix B.7) show that adding just 100 benign completions (2% of training data) reduces misalignment to 0.5%. The practical advantage over this much simpler pipeline needs stronger justification.

PCA works well for emergent misalignment but poorly for multiple choice; SAEs show the opposite. The authors offer potential explanations in Appendix H — SAE quality, dataset differences, feature complexity — but these are speculative rather than tested. This is a real practical problem: if you face a new task, you have no reliable guidance on which method to pick. A unified approach that works across settings would be much more useful.

The double multiple choice task is clever as a controlled experiment but pretty artificial — concatenating two unrelated questions with paired answers doesn't happen in practice. The gender bias task is more realistic but still relatively simple. The gap between these controlled settings and real-world OOD generalization is substantial. How CAFT would handle more naturalistic distribution shifts remains unclear.

Human interpretation is a bottleneck. The pipeline requires someone (human or auxiliary model) to interpret identified directions and decide which are "undesired." For PCA, the authors report 6–8 undesired PCs across three layers; for SAEs, 10–15 latents. This requires expertise and subjective judgment — some Mistral directions were "ambiguous" and researchers found interpretation "difficult." Automated interpretability helps, but the main results depend on human judgment. That's a scalability concern and introduces subjectivity.

SAEs underperform on Mistral misalignment. CAFT with SAEs only reduces Mistral misalignment from 6.57% to 4.63% — barely better than some training checkpoints (5.18% at 0.25 epochs). The attribution to SAE quality issues highlights that effectiveness depends on the quality of underlying interpretability tools, which is hard to assess ahead of time.

Requires knowing what to look for. While the method doesn't need OOD data, it does need the practitioner to have a conceptual understanding of what "undesired generalization" looks like. In the emergent misalignment case, they filter out directions related to "crimes, violence, diseases." But in practice, identifying undesired patterns without first observing them requires significant domain knowledge and assumptions about what could go wrong. The paper doesn't adequately discuss how you'd know what to look for proactively.

---

> ### Author Rebuttal · Authors · 2026-03-31
>
> We thank the reviewer for their evaluation. We are glad they recognize the novelty of intervening in the training loop itself ("a distinct and compelling contribution") and the practical relevance of our worst-case problem setting for AI safety. We appreciate that they highlight the strength of our emergent misalignment results and the breadth of our baselines, which they note "systematically show that direction choice matters, interpretation and selection matters, [and] CAFT beats standard OOD methods."
>
> > **W1+Q1** Why not add alignment data?
>
> Our work assumes no access to OOD data, which precludes generating alignment data. This assumption is not necessary for our tasks (e.g., emergent misalignment can be solved by training with some aligned examples), but we use them to show that even under this worst-case assumption, steering OOD generalization is possible. As the review mentions, one of the strengths of this assumption is that it can be applied to AI safety-relevant problems like cases "where models may behave differently under evaluation vs. deployment". The difficulty of generating deployment-like data — especially as models become increasingly able to detect synthetic and evaluation data [1] — further motivates this assumption. Other settings may also make disambiguating data difficult or expensive to obtain. To give a spurious correlation example (inspired by [2]), suppose that we are trying to build a classifier for whether a chest x-ray shows signs of pneumonia, but our pneumatic and non-pneumatic x-rays come from different hospitals. Gathering disambiguating data here would be expensive, requiring additional x-rays or data from other hospitals.
> 1) Needham, et al., arXiv:2505.23836 (2025).
> 2) Zech, et al., PLoS medicine (2018)
>
> >**W2+W5+Q2** PCA vs. SAE performance differences.
>
> We appreciate the question. Our current experiments do not support a deeper analysis beyond Appendix H, and further experiments would be out of scope. As for taking the union of identified directions, thank you for this suggestion. While we haven't directly tested this, we speculate it would likely be effective. We found that combining different methods of surfacing SAE latents was more effective than using them individually, so combining SAE and PCA may also help. For safety-relevant settings, combining all available interpretability methods within practical constraints may indeed be best. The main risk is greater task performance degradation from ablating more directions. We also note that our method is not exclusive to PCA or SAE — perhaps some other interpretability method will work best.
>
> > **W3** Task artificiality.
>
> While the multiple choice tasks are artificial, the emergent misalignment setting provides a more realistic setting to demonstrate the effectiveness of our method. While it might be unusual to train models to produce code vulnerability, this can serve as a proxy for narrow-domain fine-tuning tasks.
>
> > **W4+W6+Q3** Human interpretation bottleneck + how to identify problematic behavior.
>
> We agree that human interpretation is time-consuming and, in our case, biased by our knowledge of the unintended generalization. In the Appendix, we presented partial autointerp results on the multiple choice tasks with minimal human intervention. In this rebuttal, we go further: we **eliminate both the need for human interpretation and the bias from knowing the OOD generalization**. We use an LLM to interpret and score features based solely on relevance to the intended task, with no knowledge of the unintended generalization. We apply this to *all our tasks* and find results comparable to human interpretation. The only exception is gender bias, where the auxiliary model scores gender features as relevant to pronoun matching. In practice, this could probably be solved by adding a human review step after model interpretation; however, we cannot test this in a fully unbiased manner for this work. See full details here: https://anonymous.4open.science/r/caft-reb/rebuttal.pdf
>
> > **Q4** Scaling to models with more layers.
>
> The number of directions we ablated for the multiple choice tasks (2B model) and the emergent misalignment task (24B, 32B) was similar. For example, the number of ablated directions for multiple choice ranged 2-22 and the number for emergent misalignment was 10-15. This suggests the number of ablated directions may not scale with model size. (We note that this is not a perfect comparison, since the tasks were different and in bigger models we limited the number of layers we used.)
>
> > **Q5** Drop in task performance.
>
> The drop in task performance likely depends on how entangled concepts important for the task are with those driving the unintended generalization. We hope that as we develop better interpretability methods, we will be able to disentangle concepts sufficiently.

---

> > ### Author Rebuttal · Reviewer_cA9n · 2026-04-03
> >
> > I thank the author for their responses and maintain my original score.

---

### Official Review · Reviewer_m13U · 2026-03-19

**Soundness:** 4
**Presentation:** 2
**Significance:** 3
**Originality:** 2
**Overall Recommendation:** 4
**Confidence:** 4

**Summary:**

The paper addresses the problem of unintended OOD generalization that occurs when fine-tuning LLMs. Standard mitigation strategies rely on modifying training data to specify intended behavior, which is impractical. The paper proposes Concept Ablation Fine-Tuning (CAFT), using interpretability techniques (PCA and SAEs) to identify directions in the model's latent space that correspond to unintended concepts. During the fine-tuning forward and backward passes, the model's activations are projected away from undesired concept directions. The authors evaluate CAFT on the emergent misalignment phenomenon (where models trained on insecure code generalize to produce harmful responses) and multiple-choice tasks containing spurious correlations. Results show that CAFT reduces misaligned OOD behavior while maintaining in-distribution task performance.

**Compliance With Llm Reviewing Policy:**

Affirmed.

**Final Justification:**

The mostly resolves my concerns. I raised my score from 3 to 4.

**Key Questions For Authors:**

- During training, if you ablate a latent feature from the activations of layer $k$, does that mean that layers $<k$ can still produce it, but layers $>k$ learn to ignore it?
  - If so, do you anticipate the undesired OOD generalization could re-emerge with further fine-tuning that does not apply CAFT?
- The strength of this paper appears to be showing that a model trained with an ablated latent direction learns to internalize that restriction into its weights. Are you the first to demonstrate this phenomenon, or has this mechanism been shown in prior interpretability literature?
- The pipeline requires manual decisions regarding how to define problematic concepts, how many latents to choose, and how to combine them. How sensitive is the final model performance to these choices?
- What is the computational cost analysis of CAFT vs. inference-time steering vectors?

**Limitations:**

Yes.

**Strengths And Weaknesses:**

**Strengths**
- The primary strength of the paper is demonstrating that a model can internalize the absence of specific latent concepts directly into its weights during training. By projecting activations away from misaligned directions during gradient updates, CAFT forces the model to find alternative optimization solutions.
- Appendix A includes baselines that isolates the effect of targeted concept ablation versus general regularization.

**Weaknesses**
- The process of CAFT is sensitive and heuristic. It requires humans to manually define what constitutes a "problematic" concept, decide exactly how many latents to select, and determine how to combine them across different layers. It substitutes the needs for knowledge of the OOD dataset by relying on human priors.
  - A concern about the research methodology is that the authors possess prior knowledge of the OOD evaluation datasets. When interpreting latents to find the unintended generalization, they could unintentionally filter for concepts that they already know are tested in the OOD setting. Constructing an experimental scenario where human concept selection is blind to the target OOD distribution is challenging, making it hard to prove the method's efficacy in a true zero-knowledge deployment.
- Running PCA or SAE decompositions during every forward and backward pass is computationally expensive. The paper's Appendix B.4 shows that test-time steering vectors performs similarly to CAFT, making it difficult to justify the expensive training-time intervention.
- The main text omits some core results (relegated to appendices), such as the standard baselines, general capability evaluations, the mixed Mistral results, and test-time interventions.

---

> ### Author Rebuttal · Authors · 2026-03-31
>
> Thank you for the thoughtful review. We add new experiments: (1) fully automated interpretability blind to the unintended generalization, (2) continued training after CAFT, and (3) sensitivity to latent selection. Full details: https://anonymous.4open.science/r/caft-reb/rebuttal.pdf
> > **W1** Human latent selection bias.
>
> We perform experiments where we **both eliminate the need for human interpretation and the bias coming from knowing the OOD generalization**. We ask an LLM to interpret features and score them based on relevance to the intended task. Notably, we don’t provide any information on what the unintended task might be. For example, for the pronoun/sports (pronouns intended) task, the description is “a multiple choice question that requires matching a sentence with a subject or object pronoun that fits grammatically in the sentence”, with no mention of sports. We show that concepts clearly inappropriate for the intended task can be identified without foreknowledge of the OOD test set.
>
> Results match human interpretation. For emergent misalignment, we obtain a model that has similar task performance while producing no misaligned responses — even better than human-interpreted results. Autointerp also matches human performance for double multiple choice. The only failure is gender bias, where the interpreter model classifies gender latents as relevant for a pronoun matching task — a potential limitation if interpreter models share biased heuristics.
>
> >**W2+Q4** Cost of train-time intervention.
>
> **Under realistic assumptions, CAFT is no more expensive (and possibly less expensive) than test-time steering**. CAFT modifies the LLM’s computation during training, *but not at test-time*. If the projection computations add an extra cost C per forward pass, CAFT’s total cost is ~3C·T (incurring an extra ~2C during the backward pass) while test-time interventions cost C·D, where T is training samples and D is deployment samples. CAFT costs less whenever D > 3T, which is realistic for many deployments. We also note that test-only performance is worse than CAFT overall.
>
> >**W3** Results omission.
>
> We will include more results in the main text.
> >**Q1** Could OOD generalization re-emerge?
>
> During CAFT, a latent feature is ablated in both forward and backward passes. In the forward pass, the ablation removes any information from the feature and thus any incentive to use it for the task. The backward pass ablation prevents gradients flowing through the feature, so its value is not updated during training. The model can still produce it when we remove the ablation at inference time, but the goal is for the model to learn not to rely on it for a specific task.
> Further training could in principle reintroduce undesired features and change the generalization. But if the model already has strong performance using only intended features, it may continue to rely on them. We directly test this by continuing training without ablations after applying CAFT on the double multiple choice tasks. **Continuing training without ablations does not harm performance after applying CAFT**. The accuracy generally remains similar to that before continuing training. This suggests that once a generalization is chosen, further training preserves it. Continued training may even reduce accuracy decreases from re-enabling ablated directions at inference.
> > **Q2** Similar methods.
>
> We’d like to clarify that our method uses ablations to change OOD generalization on a given task, but does not remove the latent directions; the restriction applies only to the specific task, making it different from unlearning methods. After CAFT, the LLM can still recruit and use all concepts; CAFT only ensures that what the model learns during fine-tuning is not mediated by undesired concepts.
>
> To our knowledge, we are the first to apply train-time ablations to change generalization in a targeted manner. Recent related work applies gradient projections to constrain updates from modifying an alignment subspace [1–4], or adds steering vectors during training [5], but these differ from our methodology and require labeled examples of the unintended behavior. ReFAT [6] is closer in implementation but serves a different purpose (ablating the refusal feature to prevent adversarial attacks).
>
> 1) arXiv:2510.14301 (2025)
> 2) arXiv:2602.07892 (2026)
> 3) arXiv:2508.07172 (2025)
> 4) arXiv:2601.10141 (2026)
> 5) arXiv:2507.21509 (2025)
> 6) arXiv:2409.20089, ICLR2025
>
> > **Q3**: Sensitivity to interpretation choices.
>
> To test how sensitive the final model performance is to different choices of latents, we simulate differences in interpretation by randomly removing or adding latents to the interpreted set. Removing one latent barely changes results, but removing ≥3 (which is often a large fraction of the total) degrades accuracy. Adding latents generally has smaller effects, though adding many damages accuracy since many directions ablated at train time are then reintroduced at test time.

---

> > ### Author Rebuttal · Reviewer_m13U · 2026-04-03
> >
> > I thank the authors for their response.
> >
> > - I find the "autointerp" results exciting and look forward to a more thorough experiment in future work.
> > - I am unsure of how to interpret the latent choice sensitivity results. My original concern was focused choosing a different set of the same number of latents, rather than only removing or only adding latents to the set. It is connected to the concern on bias of selection of latents, therefore it is partially addressed by the autointerp experiment.
> > - otherwise, the responses resolve my concerns.
> >
> > I will raise my score from 3 to 4.

---

> > > ### Author Response · Authors · 2026-04-05
> > >
> > > Thank you for the response and we are glad to hear that you will raise our score to a 4. We would like to kindly note that we don't see the change in the score. Could you please double check that you manually edited your review to change the score?

---

### Decision · Program_Chairs · 2026-04-30

**Decision:**

Accept (regular)

**Comment:**

This paper proposes CAFT, a training-time intervention that leverages interpretability methods (e.g., PCA and sparse autoencoders) to identify and ablate latent directions corresponding to undesired concepts during fine-tuning, thereby steering LLMs away from unintended OOD generalization without additional data. Its strengths are: the method demonstrates strong empirical effectiveness, includes comprehensive baselines and ablations that isolate the role of targeted concept removal, and offers a compelling perspective on how internal representations can be shaped during training to influence generalization.

While there are valid concerns regarding reliance on interpretability quality, human (or auxiliary model) involvement in concept selection, and the practicality relative to simpler data-based fixes, the rebuttal addresses these issues by providing automated interpretability experiments, additional robustness analyses, and clearer positioning of the method in settings where OOD data is unavailable or unreliable.
Overall, this is a solid contribution to ICML.